# High-Throughput Sequencing and Expression Analysis Suggest the Involvement of *Pseudomonas putida* RA-Responsive microRNAs in Growth and Development of Arabidopsis

**DOI:** 10.3390/ijms21155468

**Published:** 2020-07-30

**Authors:** Ram Jatan, Puneet Singh Chauhan, Charu Lata

**Affiliations:** 1CSIR-National Botanical Research Institute, Rana Pratap Marg, Lucknow 226001, India; ramjatanraj@gmail.com (R.J.); puneetnbri@gmail.com (P.S.C.); 2Academy of Scientific and Innovative Research (AcSIR), Kamla Nehru Nagar, Ghaziabad 201002, India; 3CSIR-National Institute of Science Communication and Information Resources, 14 Satsang Vihar Marg, New Delhi 110067, India

**Keywords:** IAA, miRNA, PGPR, P-solubilization, siderophores, target gene

## Abstract

Beneficial soil microorganisms largely comprise of plant growth-promoting rhizobacteria (PGPR), which adhere to plant roots and facilitate their growth and development. *Pseudomonas putida* (RA) strain MTCC5279 is one such PGPR that exhibits several characteristics of plant growth promotion, such as P-solubilization, and siderophores and IAA production. Plant–PGPR interactions are very complex phenomena, and essentially modulate the expression of numerous genes, consequently leading to changes in the physiological, biochemical, cellular and molecular responses of plants. Therefore, in order to understand the molecular bases of plant–PGPR interactions, we carried out the identification of microRNAs from the roots of *Arabidopsis* upon *P. putida* RA-inoculation, and analyses of their expression. MicroRNAs (miRNAs) are 20- to 24-nt non-coding small RNAs known to regulate the expression of their target genes. Small RNA sequencing led to the identification of 293 known and 67 putative novel miRNAs, from the control and RA-inoculated libraries. Among these, 15 known miRNAs showed differential expression upon RA-inoculation in comparison to the control, and their expressions were corroborated by stem-loop quantitative real-time PCR. Overall, 28,746 and 6931 mRNAs were expected to be the targets of the known and putative novel miRNAs, respectively, which take part in numerous biological, cellular and molecular processes. An inverse correlation between the expression of RA-responsive miRNAs and their target genes also strengthened the crucial role of RA in developmental regulation. Our results offer insights into the understanding of the RA-mediated modulation of miRNAs and their targets in *Arabidopsis*, and pave the way for the further exploitation and characterization of candidate RA-responsive miRNA(s) for various crop improvement strategies directed towards plant sustainable growth and development.

## 1. Introduction

Plant growth-promoting rhizobacteria (PGPR) are associated with plants by colonizing their roots [1,2]. Various studies showed that PGPR have numerous plant growth-promoting (PGP) attributes, such as 1-aminocyclopropane-1-carboxylate (ACC) deaminase activity, indole acetic acid (IAA) and siderophore production, and P-solubilization, which help in promoting seed germination growth and the yield of plants, and also defense responses against pathogens [1,3,4,5,6,7]. Various PGPR, including *Pseudomonas* spp., have been widely used as biocontrol agents against different phytopathogens due to the production of multiple antimicrobial compounds, and in the bioremediation of environmental pollutants [8,9,10,11]. Further, PGPR have also been reportedly used as biofertilizers, due to their ability to enhance crop yields and soil fertility through the nutrient cycle between the plant roots, soil and microbes [12,13,14,15]. Modulations of multiple gene expression and changes in plant physio-chemical, cellular and molecular pathways were also due to complex interactions between plants and PGPR [6,16,17,18]. MicroRNAs (miRNAs), which are 20- to 24-nt non-coding small RNAs, regulate the expression of their target genes either by the cleavage or translational repression of target genes [19,20,21,22]. miRNAs are known to regulate plant growth, development and metabolism by controlling numerous cellular, biological and molecular processes [22,23]. Moreover, miRNAs are also reported to play very crucial regulatory roles in numerous biotic and abiotic stresses, such as drought, salt, heat, cold and CO_2_, and in response to various pathogens in different plants [24,25,26,27]. Recent studies have also shown that miRNAs perform a very critical role in plant growth and development, and in stress responses during various plant–microbe interactions [7,17,28,29]. In *Medicago truncatula*, the transcription factor *MtHAP2-1* plays an important role in the development of root nodules, and the expression of *MtHAP2*-1 was regulated by miR169a [30]. miR166 was reported to regulate the expression of an *HD-ZIPIII* transcription factor involved in root and nodule development [31]. The expression of miR172 regulated the nodulation of soybeans by targeting the *AP2* transcription factor, *Nodule Number Control1*, which suppresses the expression of *ENOD40* gene (an early nodulin gene) [32]. The expression of *mannosyl-oligosaccharide 1,2-alpha-mannosidase* and *rhizobium-induced peroxidase 1* (*RIP1*)-like peroxidase gene, regulated by gma-miR2606b and gma-miR4416, suggested that these miRNAs regulate the expression of their targets during nodulation in soybean roots upon *Bradyrhizobium japonicum* inoculation [33]. miR2111 translocated shoot-to-root in order to regulate the expression of a symbiosis suppressor TOO MUCH LOVE in the roots, leading to enhanced symbiotic susceptibility and nodulation in *Lotus japonicus* during *Mesorhizobium loti* infection [34]. Recently, it was shown that *Bacillus velezensis* FZB42 triggers induced systemic resistance (ISR) via the regulation of miRNAs and its targets, and enhances the defense response in maize against *Bipolaris maydis* infections [35].

One of the recent studies also reported the altered expression of miRNAs, as well as their target mRNAs, against the defense response in the model plant *Arabidopsis thaliana* upon *B. velezensis* FZB42 inoculation [36]. However, the involvement of miRNAs in plant growth and development upon exposure to PGPR still remains largely unexplored even in *A. thaliana.* PGPR-mediated miRNA modulation provides a unique strategy for regulating gene expression, and therefore, candidate miRNAs may be exploited as the key targets for enhancing plant agronomical characteristics, as well as for understanding their role in plant growth and development regulation. *Pseudomonas putida* (RA) strain MTCC5279 is a PGPR which has been reported to accelerate various beneficial aspects in host plants, such as nutrient uptake, growth and development, hormone production and overcome/enhance defense responses against various stresses in plants [17,37,38,39]. We have very recently shown extensive alterations in the expression profiles of conserved and novel miRNAs in RA-inoculated chickpea plants during drought stress [18]. Further, we also demonstrated that RA-inoculation results in the regulation of miRNAs, and its target genes’, expression in drought and salt stress mitigation in chickpea plants [17]. Encouraged by these findings, we were motivated to perform small RNA (sRNA) profiling in the model plant *Arabidopsis thaliana* (Col-0) in response to RA-inoculation, so as to understand PGPR-mediated developmental processes. Therefore, this study aimed to identify RA-responsive miRNAs from the roots of *Arabidopsis* using genome-wide small RNA sequencing, and to determine their function in improving the growth of host plants. In addition, the transcript expression patterns of the targets for identified miRNAs were also examined. The results obtained from the present study offer insights into the roles of RA-responsive miRNAs, and also provide a platform for further investigations into miRNA-mediated complex regulatory processes for growth and development in *Arabidopsis*. To determine the key functions of miRNAs in *Arabidopsis* upon inoculation with *P. putida*, a total of six sRNA libraries, three each from control and RA-inoculated roots, were constructed and sequenced.

## 2. Result

### 2.1. P. putida RA Improves Plant Growth by Modifications of Lateral Branches and Root Architecture

To understand whether *Arabidopsis* acts as a host plant and interacts with RA, phenotypic changes were evaluated in *Arabidopsis* plants. Numerous plant growth parameters, including root and shoot length, and number of lateral branches upon inoculation with RA-suspension culture, were then recorded (Appendix A). Further, RA-inoculated plants showed higher values concerning the fresh and dry weights of their roots and shoots, as well as in all measured parameters, as compared to the control (Appendix A).

### 2.2. Summary of sRNA Sequencing

Small RNA profiling was carried out to identify RA-responsive miRNAs from *A. thaliana* root samples, which were subjected (or not) to *P. putida* RA-inoculation. All of the six libraries of root samples, namely control_1, control_2, control_3, RA_1, RA_2 and RA_3, produced an average of 16.47 million reads, within the range of 12.24–29.76 million for each individual sample. After filtering by sequence length of >16 nt to <40 nt, and removal of the 3′ adaptor, redundant reads, tRNA, rRNA, snRNA and others, an average of 1.38 million valid and high-quality sequences were taken for further study from all six libraries. Details of the miRNA statistics are given in Table 1.

Size selection of sRNA reads suggested that high-quality sequencing results were obtained from all six libraries (Figure 1). Evaluation of the sRNA size revealed that the largest fraction (23.45%) was represented by 24-nt long sRNAs across both samples, representing the abundant presence of endogenous siRNAs. Almost 60% of the sRNAs were in the range of 21–24 nt, as predicted.

### 2.3. Identification and Analysis of Known and Novel miRNAs

After processing, 293 known and 67 putative novel miRNAs were predicted from the control and RA-inoculated root tissue samples (Appendix A). About 238, 216 and 235 conserved miRNAs from the control_1, control_2 and control_3 libraries, and 234, 256 and 233 conserved miRNAs from RA_1, RA_2 and RA3 libraries were identified, respectively (Appendix A). Nearly 87% of the known miRNAs were common, however 4% and 9% unique miRNAs were observed in the control and RA-inoculated libraries, respectively. Totals of 13, 11 and 27 conserved miRNAs were predicted from control_1, control_2 and control_3 libraries, and 16, 18 and 15 putative novel candidate miRNAs were predicted from RA_1, RA_2 and RA_3 libraries, respectively (Appendix A). A total of 12 putative novel miRNAs were common, while 29 and 26 putative novel miRNAs were unique to control and RA-inoculated libraries, respectively. Approximately 49% of the miRNAs were obtained from control samples, while the remaining 51% were from RA-inoculated samples. Out of 360 miRNAs, ~74% were found in both libraries, while ~11% and 15% were from the control and RA libraries, respectively (Figure 2a). The length and nucleotide composition analysis of the predicted miRNAs in this study revealed the highest occurrence of 21-nt long miRNAs (64%), with 5′-uridine as the most abundant nucleotide (Figure 2b).

Depending on sequence alignment, all the known miRNAs were grouped into different miRNA families. Out of 293, 107 miRNAs have not shown alignment with another miRNA, while the remaining 186 miRNAs could be grouped into 69 miRNA families. The counts of miRNAs were in the range 2–9, with the highest count for the miR156 family. A graphical representation of the miRNAs’ distribution in different families is shown in Figure 3.

### 2.4. miRNA Targets and Their GO Assessment

The prediction of target mRNAs for identified miRNAs is an important step in evaluating the regulatory functions of miRNAs. The psRNATarget server was used to predict 28,746 and 6931 putative target mRNAs from the *Arabidopsis* genome, for 180 conserved and 62 putative novel miRNAs, respectively (Appendix A). Of the 28,746 and 6931 predicted target mRNAs, 19,138 and 5687 have unique target mRNA IDs for both known and putative novel miRNAs, respectively. However, the target mRNAs could not be annotated for the remaining 113 known and 5 putative novel miRNAs, due to the stringent target prediction criteria.

GO assessment revealed that the predicted targets were involved in numerous biological processes, cellular components and molecular functions (Appendix A). GO assessment indicated that most of the target mRNAs participated in biological process-related terms (5%), followed by regulation of transcription (4%), transcription (3%) and phosphorylation-related terms (3%) in the subcategory of biological processes (Figure 4). In the subcategory of cellular components, the highest percentage of mRNAs represented membrane-related terms (14%), followed by integral components of membranes (12%) and nucleus-related terms (9%) (Figure 4). In the molecular function subcategory, target mRNAs primarily belonged to protein binding terms (9%), followed by transferase activity (5%), ATP binding (5%) and nucleotide binding-related terms (4%) (Figure 4).

### 2.5. Differential Expression Analysis of miRNAs and Their Validation

To recognize RA-responsive miRNAs from *Arabidopsis* upon RA-inoculation, DESeq was used to calculate the expression of 293 known and 67 putative novel miRNAs. Significant differential expression was observed for 15 known miRNAs in RA-inoculated plants, in comparison to the control (Appendix A). Furthermore, nine miRNAs were found to be significantly up-regulated, and six miRNAs exhibited significant down-regulation in RA-inoculated plants, as compared to the control (Figure 5a). However, no significant differential expression of putative novel miRNAs was detected upon RA-inoculation (Appendix A). Stem-loop quantitative real-time (SL-qRT) PCR was carried out for 15 differentially expressed miRNAs in order to verify the sRNA sequencing data. Similar expression patterns could be observed for 14 miRNAs (Figure 5a).

### 2.6. Expression Analysis of miRNAs Targets

Analysis of the expressions of 14 targets was carried out to correlate the expression profiles of miRNAs and their corresponding targets (Figure 5b). Therefore, qRT-PCR was carried out for *CUP-SHAPED COTYLEDON1* [*CUC1* (AT3G15170.1)—target of miR164a], *Natural resistance-associated macrophage protein 1* [*NRAMP1* (AT1G80830.1)—target of miR167b], *Salt-inducible zinc finger 1* [*SZF1* (AT3G55980.1)—target of miR390b-3p], *TRANSPORT INHIBITOR RESPONSE 1* [*TIR1* (AT3G62980.1)—target of miR393a-5p], *Growth-regulating factor 1* [*GRF1* (AT2G22840.1)—target of miR396a-5p], *Brassinosteroid-related acyltransferase 1* [*BAT1* (AT4G31910.1)—target of miR398a-3p], *Succinate dehydrogenase 5* [*SDH5* (AT1G47420.1)—target of miR822-5p], *LATERAL ORGAN BOUNDARIES DOMAIN 16* [*LBD16* (AT2G42430.1)—target of miR837-3p], *Glycerol-3-phosphate acyltransferase 4* [*GPAT4* (AT1G01610.1)—target of miR8175], *Isocitrate dehydrogenase VI* [*IDH-VI* (AT3G09810.1)—target of miR164c-5p], *TCP* family transcription factor 4 [*TCP4* (AT3G15030.1)—target of miR319c], Tetratricopeptide repeat (TPR)-containing protein [*TSK* (AT3G18730.1)—target of miR4245], *GRF1*-interacting factor 2 [*GIF2* (AT1G01160.1)—target of miR837-5p] and *Nitrate transporter 1.7* [*NRT1.7* (AT1G69870.1)—target of miR838]. However, the target for miR870-5p could not be predicted due to the stringent target prediction criteria. Nine miRNAs, namely miR164a, miR167b, miR390b-3p, miR393a-5p, miR396a-5p, miR398a-3p, miR8175, miR822-5p and miR837-3p, demonstrated an inverse expression as compared to their corresponding targets, while the targets of the remaining five miRNAs (miR164c-5p, miR319c, miR4245, miR837-5p and miR838) showed basal expression.

## 3. Discussion

Understanding the vital roles of PGPR in plant growth and development has become imperative, owing to their important applications in emerging crop production technologies as an alternative strategy to breeding and genetic engineering. Several studies have reported that PGPR have been involved in the response to abiotic and biotic stresses, and in various growth and development processes [35,40,41]. Recently, expressions of miRNAs have been reported to be modulated by PGPR during the plant–PGPR interactions [17,18,35,36]. Nevertheless, very limited evidence of PGPR-responsive miRNAs and their target genes is available as regards plant growth and development. Therefore, high throughput sequencing was carried out to identify *P. putida* RA-responsive miRNAs, and their target mRNAs related to *Arabidopsis* growth and development. It was previously demonstrated that *Arabidopsis* acts as a host plant for several PGPR [3,42,43]. We have also established that RA-inoculation has led to an increase in the fresh and dry weights of the roots and shoots of *Arabidopsis*. In addition, the number of lateral shoots was also found to be increased as a result of RA-inoculation. Taken together, the interactions between *Arabidopsis* and *P. putida* RA may act as an appropriate model system for studying beneficial plant–PGPR relations. Signaling and translocation of miRNAs during nutrient starvation responses and specific developmental events has been reported in different plant species in various studies [34,44,45,46,47]. Although we have discerned increased growth in the above-ground parts of *Arabidopsis* upon RA-inoculation, future research still needs to explore the translocation of miRNAs and their involvement in the epigeous growth and development of plants during PGPR interactions. In the rhizosphere, PGPR adhere to plant roots and contribute to complex processes, leading to root system architecture modification, improved plant nutrition and changes to the physiology and growth of the whole plant [2]. Considering this, the identification of RA-responsive miRNAs from roots was taken up in this study.

High-throughput sequencing provided an average of more than 16 million reads of sRNA and bioinformatics analyses, used to predict known and novel miRNAs from the control and RA-inoculated libraries. Length distribution demonstrated that sRNA reads containing 24-nt sRNAs were highly dominant with regards to length, followed by 21-nt sRNA, and this type of distribution is significantly assessed by various DCL and AGO associations [48,49]. Our results were in accordance with earlier reports from different plants, which showed that sRNAs containing 24 nt were highly abundant in the sRNA reads [48,49,50]. In our study, approximately 60% of the sRNAs were in the range of 21–24-nt, conforming to the DCL cleavage, as shown in previous reports (Figure 1) [51,52]. miRNA length and nucleotide distribution have suggested that the presence of 21-nt miRNAs with abundant 5′-uridine residues is a unique property of AGO1-associated DCL1 cleavage (Figure 2b) [51,53]. miRNA length distribution is an important step in the alignment with RNA-induced silencing complex (RISC), which results in the degradation of the target or inhibition of its translation, based on the complementarity of miRNA with mRNA in regulating the gene expression.

The predicted 293 known miRNAs from the control and RA-inoculated root samples were clustered into 69 miRNA families. The most frequent occurrence of the miR156 family suggested their significant role in the growth and development of *Arabidopsis*. Plant growth and development are regulated by complex gene regulatory networks. miRNAs have previously been demonstrated to regulate target expression through the miRNA-mediated translational repression/cleavage of mRNAs in plants [54,55]. Therefore, the prediction of targets was performed in order to identify the regulatory roles of miRNAs. Subsequently, GO analysis showed their involvement in various biological processes, cellular components and molecular functions (Figure 4). These findings suggested that the identified miRNAs regulate various functions in *Arabidopsis* during RA-inoculation.

SL-qRT-PCR analysis indicated the most significant up-regulation for miR393a-5p (2.58-fold), and the most significant down-regulation for miR319c (−2.75-fold), upon RA-inoculation. However, miR164c-5p exhibited basal expression in RA-inoculated plants. Further, patterns of miRNA expression identified through SL-qRT-PCR were largely in accordance with the small RNA sequencing results, validating the reliability of RNASeq data in our analysis.

*CUC1* was predicted to be the target of miR164a in our study, and an inverse correlation between the expressions of the two was also observed. *CUC* belongs to the *NAC* TF superfamily, and is negatively regulated by the miR164 involved in tissue differentiation and the establishment of axillary meristems [56,57,58]. In addition, the miR164 regulatory framework of *CUC1* and *CUC2* is responsible for leaf development as well as appropriate formation of organ boundaries during the development of flowers [59,60,61]. Significantly up-regulated miR164a led to the down-regulation of *CUC1* in RA-inoculated *Arabidopsis* plants, raising the possibility that the RA-mediated regulation of the miR164a-CUC1 module might be responsible for leaf development, as well as the proper formation of organ boundaries during flower development.

The *NRAMP* gene family encodes a metal transporter, and is reported to be involved in the transport of bivalent metal ion throughout the plasma membranes. *NRAMP1* has been shown to maintain iron (Fe) homoeostasis, and also act as a high-affinity transporter for manganese (Mn) and zinc (Zn) uptake in various plant species [62,63,64,65]. In a previous study, the *NRAMP1b* gene of *Brassica napus* was reported to be the target of miR167 [62]. In our study, the up-regulated expression of miR167a mirrored the down-regulation of the *NRAMP1* mRNA upon RA-inoculation. Therefore, it is assumed that modulation of the miR167-NRAMP1 module is involved in the regulation of Fe homoeostasis and Mn uptake, as well as translocation, in RA-inoculated *Arabidopsis*.

miR393 is a conserved miRNA family that targets F-box-containing genes, such as *TRANSPORT INHIBITOR RESPONSE1* (*TIR1*) and auxin-signaling F-box proteins (AFBs), in various plants [66,67,68]). miR393 plays a very crucial role in auxin signaling, and regulates numerous biological processes, such as leaf and root development, and responses to various biotic and abiotic stresses [66,67,69,70,71,72]. Besides, miR393 was also reported to play important roles in the response to nitrate availability, root system architecture, salinity stress and the inhibition of root elongation during aluminum stress [70,73,74,75]. In some of the previous studies, the miR393-TIR1 module was demonstrated to regulate crown root initiation, flag leaf inclination, tillering, seminal root development, fruit/seed set development and leaf morphogenesis [66,76,77]. In our study, miR393a-5p, a member of miR393 miRNA family was found to be significantly up regulated, and their predicted target, *TIR1*, showed significant down-regulation upon RA-inoculation. Together, our result suggested that the miR393a-5p-TIR1 module might be involved in the regulation of growth and development of *Arabidopsis* during RA-inoculation.

miR396 targets the *GRF* TF family members, and miR396-mediated regulation of the GRFs involved in the growth and development of the plants during biotic and abiotic stresses has also been reported [78,79,80]. miR396 has been found to regulate the GRF–GIF complex that plays a very crucial role in the development of the pistil and the carpel number, and in specifying the meristematic cells of gynoecia and the anthers in *Arabidopsis* [81]. Our results showed that the significant up-regulation of miR396a-5p led to the significant down-regulation of *GRF1* in RA-inoculated samples, as compared to the control. Hence, it was speculated that RA could potentially be involved in the growth and development of *Arabidopsis* by regulating the expression of miR396a-5p and *GRF1* during the interaction of RA with *Arabidopsis*.

*BAT1* has been demonstrated to play important roles in the regulation of brassinosteroids (BRs) homeostasis, via the conversion of active BR intermediates, such as typhasterol, 6-deoxocastasterone and 6-deoxotyphasterol, into inactive acylated BR conjugates [82,83,84]. BRs are naturally produced steroid hormones, and the maintenance of their homeostasis is responsible for numerous processes during the growth and development of plants. The regulation of active BRs and their homeostasis influences or has a positive impact on the production of biomass and the yield of grains in plants [82,85,86]. Therefore, *BAT1* could be a relevant target for regulating the active BRs level and homeostasis, eventually leading to increased plant productivity. In our study, a significant up-regulation of miR398a-3p was found during RA-inoculation, as compared to the control, and the opposite expression pattern was observed for their predicted target, *BAT1*. Thus, the RA-mediated regulation of miR398a-3p and *BAT1* might provide a way to improve the agronomic traits of crops, including biomass and yield, by regulating the level of active BRs and homeostasis upon RA-inoculation.

The *LBD* gene family encodes plant-specific TFs that contain the LATERAL ORGAN BOUNDARIES (LOB) domain, and are reported to play important roles in the lateral organ development and metabolic processes in higher plants [87,88,89]. In earlier reports, *LBD* has been shown to regulate the pollen development, plant regeneration, photomorphogenesis and response to pathogens, as well as some specific developmental processes of non-model plants, including poplars and legumes [90,91,92,93,94]. In *Arabidopsis*, *AtLBD16*, *AtLBD17*, *AtLBD18* and *AtLBD29* were reported as important regulators of callus initiation in different organs, and they regulate lateral root formation [90]. Additionally, *LBD16* was reported to be responsible for the initiation of lateral roots and for determining the asymmetry in the founder cells of lateral roots prior to cell division [95,96]. In this study, *LBD16* was predicted as the target of miR837-3p, and its expression was found to be significantly down-regulated, while those of miR837-3p showed significant up-regulation upon RA-inoculation, suggesting that the RA-mediated regulation of the miR837-3p-LBD16 module may be involved in lateral root formation in *Arabidopsis*. Further elaborate studies will help in uncovering the molecular mechanisms governing the RA-mediated regulation of miR837-3p and *LBD* in lateral organ development and other processes in plants.

Furthermore, it is well known that one target gene can be regulated by multiple miRNAs. As a result, the regulatory mechanisms between miRNAs and their target mRNAs could be more complicated. In our study, five miRNAs, namely miR164c-5p, miR319c, miR4245, miR837-5p and miR838, did not exhibit exact inverse correlations with their predicted targets (*IDH-VI*, *TCP4*, *TSK*, *GIF2* and *NRT1.7*, respectively) (Figure 5b). Therefore, we speculated that these five miRNAs may have a lesser impact on their targets, and the expression of the target mRNAs could be influenced by other regulators or by spatio-temporal regulation. However, more research needs to be done in order to uncover the regulatory mechanisms of these miRNAs with regard to their predicted target genes for the growth and development of *Arabidopsis* during RA-inoculation.

## 4. Materials and Methods

### 4.1. Plant Materials and Growth Conditions

*Arabidopsis thaliana* ecotype Col-0 was used to identify miRNAs and their expression profiling upon *Pseudomonas putida* RA-inoculation. *Arabidopsis* seeds were surface sterilized, vernalized for three days, sown in pots containing soilrite supplemented with Somerville and Ogren medium [97] (macronutrient: 5 mM KNO_3_, 2 mM MgSO_4_.7H_2_O, 2 mM Ca(NO_3_.7H_2_O), 50 μM Fe EDTA, 2.5 mM KPO_4_ (pH 5.5) and micronutrient: 70 mM H_3_BO_3_, 14 mM MnCl_2_.4H_2_O, 1 mM ZnSO_4_.7H_2_O, 0.5 mM CuSO_4_.5H_2_O, 0.2 mM Na_2_MoO_4_.7H_2_O, 10 mM NaCl, 0.01 mM CoCl_2_), and grown in a growth chamber for three weeks under standard growth conditions. A single isolated colony of *P. putida* RA was inoculated in NB medium and grown at 28 °C, 180 rpm, in a rotary incubator shaker overnight. When the final bacterial concentration reached ~10^7^ CFU mL^−1^, the cells were harvested at 5000 rpm for 5 min and the pellet was dissolved in sterile Milli-Q. Three-week-old *Arabidopsis* were then supplemented with liquid culture of *P. putida* RA and allowed to grow for another three weeks. Uninoculated plants were taken as control. Biological triplicates (three plants per replicate) of the control and RA-inoculated root samples were collected after three weeks of RA-inoculation, then quickly frozen in liquid nitrogen and stored at −80 °C for RNA isolation. Phenotypic data was analyzed to determine root and shoot length, number of lateral branches, and the fresh and dry weights of roots and shoots.

### 4.2. Preparation of sRNA Libraries for Sequencing and Data Analysis

Total RNA was extracted from the control and RA-inoculated root tissues of *Arabidopsis* via mirVana miRNA Isolation Kit (Ambion, Austin, TX, UAS) in three biological replicates, namely control_1, control_2 and control_3, RA_1, RA_2 and RA_3. The concentration and purity of the RNA samples was assessed by a Nanodrop spectrophotometer (Thermo Scientific, Waltham, MA, USA) and Qubit fluorometer (Thermo Scientific, Waltham, MA, USA). The integrity of RNA was also analyzed by BioAnalyzer 2100 (Agilent Technologies, Palo Alto, CA, USA) for quality check and samples with RIN ≥ 8.0 were used for sRNA library preparation and sequencing. In order to complete the genome-wide identification of RA-responsive miRNAs, three biological replicates each of the control and RA-inoculated *Arabidopsis* root tissues were utilized for generating six sRNA libraries, using a Illumina Truseq small RNA library preparation kit (Illumina Technologies, San Diego, CA, USA). The preparation and sequencing of the sRNA libraries and the data analysis was carried out by Genotypic Technology (Bangalore, India) using the Illumina NextSeq500 Single-end sequencing (75 × 1) platform (Illumina, San Diego, CA, USA). All raw data were submitted to the SRA database under the accession number PRJNA591259. The raw data in FASTQ format were generated on the Illumina platform, and srna-workbenchV3.0_ALPHA [98] was used to remove adapters and sort sRNA reads according to the size of nucleotides (>16 bp and <40 bp). For the selection of only unique reads for further analysis, contaminated and low-quality reads were eliminated.

### 4.3. Identification and Expression Analysis of Arabidopsis miRNAs

All the unique reads were aligned to the *A. thaliana* reference genome (*Arabidopsis_thaliana* TAIR10 ENSEMBL) using Bowtie-1.1.1 [99]. The aligned reads were extracted and evaluated for ncRNA (rRNA, snRNA, snoRNA and tRNA) contamination against the Rfam database (http://www.sanger.ac.uk/software/Rfam) [100], while known miRNAs were predicted from unaligned reads. Repeated miRNA reads have been made unique using the CD-HIT4 clustering approach, and thus read count has been obtained [101]. For prediction of known miRNAs, NCBI-Blast-2.2.30 [102] was used to perform the sequence homology of unaligned reads with *Arabidopsis* mature miRNAs obtained from miRbase-22 database (http://www.mirbase.org/) [103]. The precursor miRNA sequences for known miRNAs were extracted from the reference database.

Mireap_0.22b was used to predict novel miRNAs from the remaining sequence reads that did not align to any known miRNAs [104]. Further, a 100-bp flanking sequence of aligned reads on the genome was used to extract hairpin sequences, and then potential precursor secondary structures were obtained through the RNA/folding annotation tool in the Vienna RNA package (http://rna.tbi.univie.ac.at/cgi-bin/RNAWebSuite/RNAfold.cgi) [105].

For differential expression analysis of miRNAs, DESeq R package was used (https://www-huber.embl.de/users/anders/DESeq/) [106]. Variations in the reads were normalized by the library normalization method adopted from DESeq library. DESeq calculates size factor where each read count is normalized by dividing with size factor. miRNA normalized read counts of the RA-inoculated sample and those of the control sample were used for the calculation of differentially expressed miRNAs. miRNAs with ≥ 1.0 and ≤ −1.0 log_2_ fold change having <0.05 *p*-value were regarded as significantly differentially expressed.

### 4.4. Target Prediction and Their Functional Annotation

miRNAs with a copy number ≥5 were used to predict the targets of miRNAs by using psRNATarget server (http://plantgrn.noble.org/psRNATarget/) [107]. The cDNA sequence library of *Arabidopsis thaliana* from EnsemblPlants (http://plants.ensembl.org) was used for target prediction using psRNATarget server (V2, 2017 release) following the given parameters (Max Expectation cutoff: 5; Penalty for GU pair: 0.5; Penalty for other mismatch: 1.0; Weight in seed region: 1.5; Seed region: 2 ± 13 nt; # of mismatches allowed for seed region: 2; HSP size: 19; Allowing bulge on target: Yes; Penalty for opening gap: 2.0; Penalty for extending gap: 0.5; Calculating target accessibility: Yes; Max UPE: 25; Flanking length up/downstream: 17/13 nt; Translational inhibition range: 9–11 nt). miRNA-target pairs along with Gene Ontology (GO) and pathway annotation have been done. GO analysis showed that the target gene classification of known miRNAs was similar to that of the newly identified miRNAs. GO annotation was performed using AgriGO (http://bioinfo.cau.edu.cn/agriGO) [108].

### 4.5. Validation and Expression Analysis of miRNAs and Their Targets

Validation of miRNA expressions and expression analysis of their target genes were carried out according to Jatan et al. (2018) [17]. The primer pairs are given in Appendix A. Tubulin was used as an internal control for data normalization and the 2^−ΔΔCt^ method was utilized to determine the relative expressions of miRNAs and their target genes from the mean values of three independent biological and three technical replicates [109].

### 4.6. Statistical Analysis

The triplicates (100-mg root samples) were used to determine the mean and standard error (mean ± SEM) for the relative expression results of miRNAs and their target mRNAs. Significant differences among the average values of the control and RA-inoculated samples were calculated by an unpaired two-tailed *t*-test at *p* < 0.05 using GraphPad Prism software (version 5.03, San Diego, CA, USA). Graphs for the expression results were illustrated using GraphPad Prism software (version 5.03, https://www.graphpad.com/scientific-software/prism/).

## 5. Conclusions

To summarize, a genome-wide profiling of miRNAs from *P. putida* RA-inoculated and un-inoculated *Arabidopsis* roots was performed via high-throughput sequencing technology and a bioinformatics approach. Functional annotation showed that the targets of RA-responsive miRNAs from *Arabidopsis* belonged to different transcription factors and other protein coding genes associated with many aspects of plant growth and development. The expression of miRNAs being opposite to that of their targets highlighted the complexity of the RA-mediated control of miRNAs and its target network in *Arabidopsis* after RA-inoculation. Our findings suggested that the RA-mediated regulation of miRNAs and their targets during association with RA could assist in the growth and development of *Arabidopsis*, and provides a new opportunity for understanding the complexity of the RA-mediated regulation of miRNAs and their target modules in improved crop production programs.

## Figures and Tables

**Figure 1 ijms-21-05468-f001:**
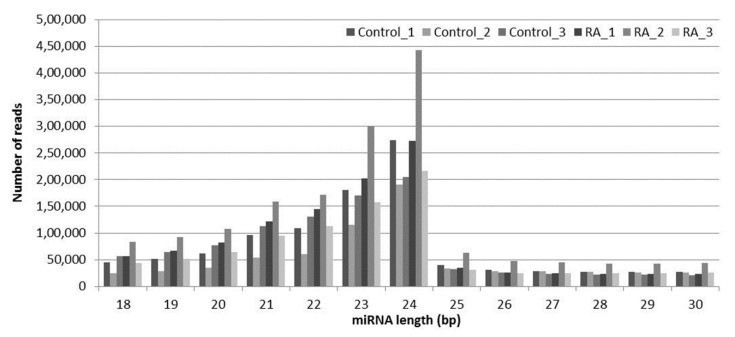
Size distribution of small RNA reads from *Arabidopsis*.

**Figure 2 ijms-21-05468-f002:**
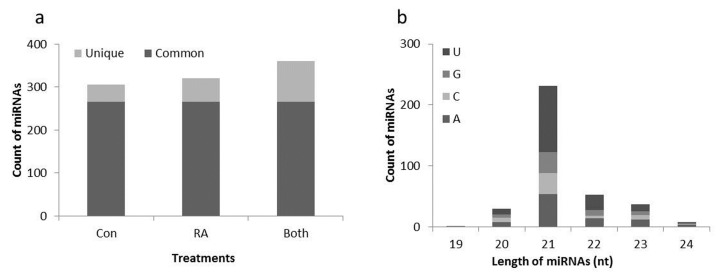
Predicted miRNAs from *Arabidopsis* roots. (**a**) Count of identified miRNAs; (**b**) nucleotide length distribution and the characterization of the 5′ end first nucleotide of miRNAs.

**Figure 3 ijms-21-05468-f003:**
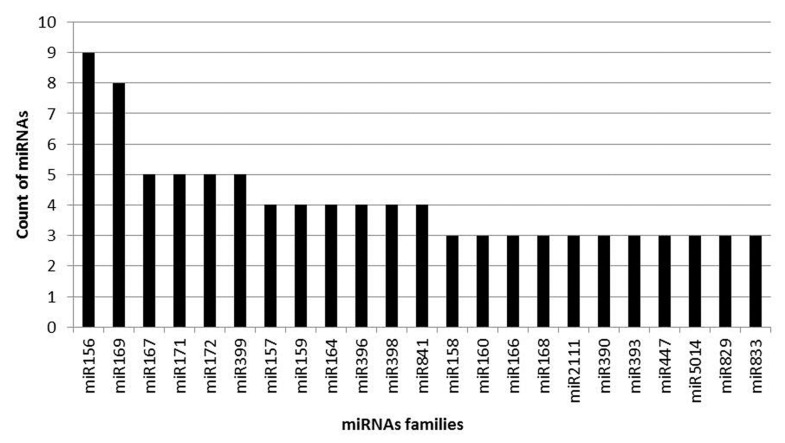
Distribution of conserved miRNA families.

**Figure 4 ijms-21-05468-f004:**
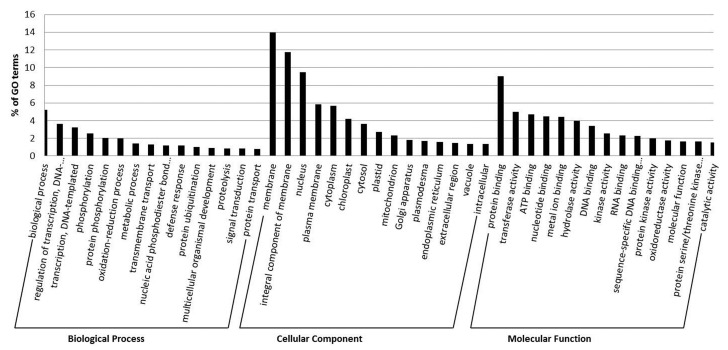
Gene ontology (GO) analysis of the predicted targets of miRNAs and most abundant GO terms from biological processes, cellular components and molecular functions.

**Figure 5 ijms-21-05468-f005:**
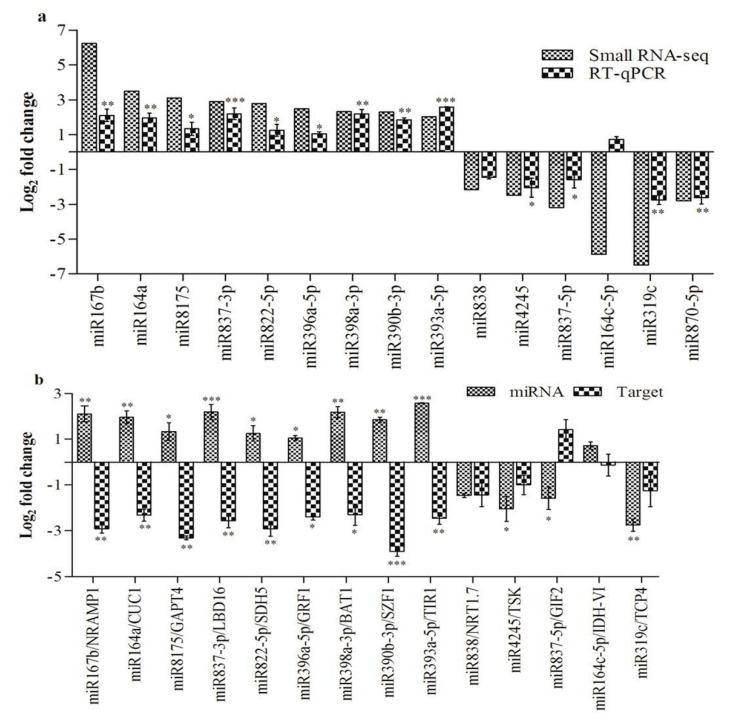
Differential expression analysis of miRNAs and target genes. (**a**) Small RNASeq and SL-qRT-PCR comparative expression analysis of miRNAs. (**b**) Comparative expression of target genes by qRT-PCR with their predicted targets. Error bars represent the mean and standard error of value obtained from three biological replicates. *Asterisks* indicate significant difference in RA-inoculated plants as compared to the control, * *p* < 0.05, ** *p* < 0.01, *** *p* < 0.001 by unpaired t test.

**Table 1 ijms-21-05468-t001:** Statistics of small RNA sequencing data from all six libraries of *Arabidopsis*.

	Control_1	Control_2	Control_3	RA_1	RA_2	RA_3
Total Reads	14,686,655	12,373,692	14,639,734	14,202,847	29,7684,85	13,162,721
Trimmed unique reads	1,319,211	986,556	1,235,562	1,404,413	2,156,182	1,215,179
Reads aligned to genome	1,174,700	846,192	1,114,548	1,270,093	1,888,067	1,100,485
rRNA	172,731	172,194	128,009	135,128	248,281	145,704
snoRNA	6578	7353	6214	7699	9254	8057
snRNA	4184	3679	4371	4609	6912	4566
tRNA	23	13	25	29	60	27
Clustered reads	584,288	408,865	542,203	614,695	863,035	526,561
Reads aligned to miRBase	4448	3467	4574	4943	7029	4108
Known miRNA unique	238	216	235	234	256	233
Reads utilized for Novel miRNA	567,702	397,267	524,730	597,830	832,030	513,748
Novel miRNA predicted	13	11	27	16	18	15

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
