# Peer review of "High-Throughput Sequencing and Expression Analysis Suggest the Involvement of Pseudomonas putida RA-Responsive microRNAs in Growth and Development of Arabidopsis"

_ijms, 2020, doi:10.3390/ijms21155468_

Round 1

Reviewer 1 Report

The manuscript is interesting and in general it seems well written. 

I have only some minor points that should be addressed to improve it. 

In the abstract the authors should already explain what the miRNAs are instead that to leave only the acronym. 

In the Introduction, I would suggest to improve a bit the part on PGPR writing some sentences more on this topic. Additionally, what about the role of miRNA on gene expression in other plant-microbe interactions? Probably the authors could add something about this point. 

In M&M, the paragraph 2.5 on validation should be improved adding for example the considered biological and technical replicates. 

Author Response

Responses to the Reviewers’ comments:

We are thankful to the reviewer for their helpful suggestions. We have incorporated the suggested changes/modifications and other required changes in track change mode in the manuscript. Please find below the responses to reviewer’s comments:

Response to Reviewer 1 Comments:

Comments and Suggestions for Authors

The manuscript is interesting and in general it seems well written. 

Response: Thanks for the appreciation.

I have only some minor points that should be addressed to improve it. 

In the abstract the authors should already explain what the miRNAs are instead that to leave only the acronym. 

Response: Thanks for the suggestion. We have now incorporated a sentence explaining the miRNAs in the revised manuscript in the lines 22-24.

In the Introduction, I would suggest to improve a bit the part on PGPR writing some sentences more on this topic. Additionally, what about the role of miRNA on gene expression in other plant-microbe interactions? Probably the authors could add something about this point. 

Response: Thanks for the suggestion. Introduction section has now been improved as suggested in the lines 43-48. Necessary modifications regarding the role of miRNA on gene expression in other plant-microbe interactions have now been done in the revised manuscript in the lines 57-69.

In M&M, the paragraph 2.5 on validation should be improved adding for example the considered biological and technical replicates. 

Response: Thanks for your valuable suggestion. Necessary modifications have been done accordingly in the revised manuscript in lines 399-400 (Now in the paragraph 4.5, because the section order has been modified according to following order: 1. Introduction, 2. Results, 3. Discussion, 4. Material and Methods, 5. Conclusions).

Reviewer 2 Report

In this study, the authors carried out identification of miRNAs from roots of Arabidopsis upon Pseudomonas putida RA-inoculation by high-throughput sequencing. For this, a total of six sRNA libraries, three each from control and RNA-inoculated roots were constructed and sequenced. In addition, the results were validated by stem-loop quantitative real-time PCR and the transcript expression patterns of the targets for identified miRNA were also examined.

Overall, the manuscript is clearly written and the experimental approach is correct. The work contributes new knowledge to the field of the transcriptomic changes in miRNA profile associated with PGPR in Arabidopsis. The paper is suitable for publication, but there are several points that need to be clarified before.

In this study, the RA-inoculated plants were grown in a growth chamber for three weeks under non-sterile conditions, then the observed effects may be affected by the growth of other microorganisms. This should be considered by the authors.

Line 159_  I consider that the sentence is not appropiate since there previous works in which the interaction between Pseudomona putida and Arabidopsis is studied, for example in the work of Srivastava et al. 2012. In this work, no significant differences in root length were observed between RA-inoculated plants and control plants, while differences were observed in this study (cnt, 15.86±0.81; RA-inoculated plants, 18.39 ±0.98), although in the photo of figure S1 no clear significant differences are observed. Could the authors justify this result? I consider that the number of plants used to analyse the different parameters is low, so the authors should increase the number of samples, both form control plants and RA-inoculated plants.

Author Response

Responses to the Reviewers’ comments:

We are thankful to the reviewer for their helpful suggestions. We have incorporated the suggested changes/modifications and other required changes in track change mode in the manuscript. Please find below the responses to reviewer’s comments:

Response to Reviewer 2 Comments:

Comments and Suggestions for Authors

In this study, the authors carried out identification of miRNAs from roots of Arabidopsis upon Pseudomonas putida RA-inoculation by high-throughput sequencing. For this, a total of six sRNA libraries, three each from control and RNA-inoculated roots were constructed and sequenced. In addition, the results were validated by stem-loop quantitative real-time PCR and the transcript expression patterns of the targets for identified miRNA were also examined.

Response: We are very thankful to you for appreciating our work.

Overall, the manuscript is clearly written and the experimental approach is correct. The work contributes new knowledge to the field of the transcriptomic changes in miRNA profile associated with PGPR in Arabidopsis. The paper is suitable for publication, but there are several points that need to be clarified before.

 In this study, the RA-inoculated plants were grown in a growth chamber for three weeks under non-sterile conditions, then the observed effects may be affected by the growth of other microorganisms. This should be considered by the authors.

Response: We are thankful for your concerns. We have already considered the above issue. Accordingly, growth chamber was fumigated with formaldehyde (40%) and potassium permanganate as well as UV light was also used for sterilization purpose before transferring the pots. The seeds were then allowed to germinate and grown for three weeks in the growth chamber. Further no visible growth was observed for other microorganisms.

Line 159_  I consider that the sentence is not appropiate since there previous works in which the interaction between Pseudomona putida and Arabidopsis is studied, for example in the work of Srivastava et al. 2012. In this work, no significant differences in root length were observed between RA-inoculated plants and control plants, while differences were observed in this study (cnt, 15.86±0.81; RA-inoculated plants, 18.39 ±0.98), although in the photo of figure S1 no clear significant differences are observed. Could the authors justify this result? I consider that the number of plants used to analyse the different parameters is low, so the authors should increase the number of samples, both form control plants and RA-inoculated plants.

Response: Thanks for raising your concern and valuable suggestion. Here, we would like to draw your attention to a recent work by the same group where they have reported that the interaction between Pseudomonas putida and Arabidopsis leads to better root architecture both in terms of enhanced number of lateral roots as well as better growth of main root (Srivastava & Srivastava 2020; https://doi.org/10.1038/s41598-020-62725-1). This reference has now been added in line 209. In figure S1, the root length of RA-inoculated plants was found to be approximately 8 cm while in the control plants root length was observed approximately 6 cm. Seven replicates of the control and RA-inoculated plants were used (other than those used for the figure S1) for root length measurement and analysis of other parameters. As far as increasing the number of plants used to analyse different parameters is concerned, we would like to state that it would not be possible at present as we are not carrying out this experiment in the growth chamber currently, and since the revised manuscript needs to be submitted within five days, a reanalysis is out of scope at present.
